# PPIases Par14/Par17 Affect HBV Replication in Multiple Ways

**DOI:** 10.3390/v15020457

**Published:** 2023-02-06

**Authors:** Kyongmin Kim

**Affiliations:** 1Department of Microbiology, Ajou University School of Medicine, Suwon 16499, Republic of Korea; kimkm@ajou.ac.kr; Tel.: +82-31-219-5072; Fax: +82-31-219-5079; 2Department of Biomedical Science, Graduate School of Ajou University, Suwon 16499, Republic of Korea

**Keywords:** peptidyl-prolyl *cis/trans* isomerase, PIN4, parvulin 14, parvulin 17, hepatitis B virus, HBV replication, HBx, HBc, core particle, cccDNA

## Abstract

Human parvulin 14 (Par14) and parvulin 17 (Par17) are peptidyl-prolyl *cis/trans* isomerases that upregulate hepatitis B virus (HBV) replication by binding to the conserved ^133^Arg-Pro^134^ (RP) motif of HBc and core particles, and ^19^RP^20^-^28^RP^29^ motifs of HBx. In the absence of HBx, Par14/Par17 have no effect on HBV replication. Interaction with Par14/Par17 enhances the stability of HBx, core particles, and HBc. Par14/Par17 binds outside and inside core particles and is involved in HBc dimer–dimer interaction to facilitate core particle assembly. Although HBc RP motif is important for HBV replication, R133 residue is solely important for its interaction with Par14/Par17. Interaction of Par14 and Par17 with HBx involves two substrate-binding residues, Glu46/Asp74 (E46/D74) and E71/D99, respectively, and promotes HBx translocation to the nucleus and mitochondria. In the presence of HBx, Par14/Par17 are efficiently recruited to cccDNA and promote transcriptional activation via specific DNA-binding residues Ser19/44 (S19/44). S19 and E46/D74 of Par14, and S44 and E71/D99 of Par17, are also involved in the recruitment of HBc onto cccDNA. Par14/Par17 upregulate HBV replication via various effects that are mediated in part through the HBx–Par14/Par17–cccDNA complex and triple HBc, Par14/Par17, and cccDNA interactions in the nucleus, as well as via core particle-Par14/Par17 interactions in the cytoplasm.

## 1. Introduction

Hepatitis B virus (HBV) infection can cause acute or chronic hepatitis, and chronic hepatitis B (CHB) infection can lead to liver fibrosis, cirrhosis, or hepatocellular carcinoma (HCC) [1]. Currently available CHB therapies cannot cure HBV infection [2,3].

HBV, the prototype virus of the family *Hepadnaviridae*, has a partially double-stranded, relaxed-circular DNA genome [1,4]. The HBV life cycle has been investigated thoroughly; however, the roles of various host proteins in HBV replication have not been characterized fully. Upon infection, the relaxed-circular DNA genome of HBV is translocated to the nucleus and converted to a covalently closed circular DNA (cccDNA). This cccDNA serves as a template for the transcription of viral RNAs, including a 3.5 kb pregenomic RNA (pgRNA) encoding the core protein (HBc) and polymerase proteins; 2.4 kb and 2.1 kb subgenomic RNAs (sgRNAs) encoding large, middle, and small HBV surface proteins (LHBs, MHBs, and SHBs, respectively); and a 0.7 kb sgRNA encoding the HBV X (HBx) protein [1,4,5]. The HBV nuclear cccDNA is organized as a minichromosome in association with histone and non-histone cellular and viral proteins, including viral HBx and HBc [6,7,8,9,10].

Proline (Pro) residue in proteins can exist in distinct *cis* and *trans* peptide bond conformations. Peptidyl-prolyl *cis/trans* isomerases (PPIases) regulate protein folding and function by twisting the backbones of target proteins through prolyl *cis/trans* isomerization [11,12]. These isomerases are involved in a wide variety of physiological and pathophysiological processes, including gene expression; signal transduction; cell differentiation; apoptosis; and viral, bacterial, and parasite infections [13,14]. PPIases are expressed abundantly in all organisms and cellular compartments [15,16].

The PPIase superfamily is subdivided into four types or families: cyclophilins, FK506-binding proteins (FKBPs), parvulins, and protein Ser/Thr phosphatase 2A activator (PTPA) [12,17]. Cyclophilins bind to the immunosuppressive drug cyclosporin A, and FKBPs are inhibited by FK506 and interact with the immunosuppressant and anticancer agent rapamycin [12]. Therefore, cyclophilins and FKBPs are referred to as immunophilins and have been studied intensely [12]. The role of PTPA is still unclear. Parvulins are involved in cellular maintenance, mitotic regulatory mechanisms, and cell proliferation [17]. The name parvulin is derived from the Latin word parvulus (meaning tiny) and was chosen based on the relatively low molecular weight (10 kDa) of the first parvulin identified, Par10 in *Escherichia coli*, which is much smaller than cyclophilins and FKBPs [18]. The human genome contains two parvulin genes, *PIN1* and *PIN4* [19,20,21,22]. The *PIN1* gene encodes the PPIase never in mitosis A-interacting 1 (Pin1), and the *PIN4* gene encodes the parvulin14 (Par14) and parvulin17 (Par17) proteins [19,20,21,22]. Several cyclophilins, FKBP37, and Pin1 play a role in HBV infection [23,24,25,26]. In addition, involvement of Par14/Par17 in the HBV life cycle has been reported recently [27,28]. However, the exact roles of Par14/Par17 in viral pathogenesis and hepatocarcinogenesis are still unclear. In this review, we discuss the interaction of Par14/Par17 with HBV proteins and cccDNA, as well as their overall effect on the HBV life cycle.

## 2. The Effects of Host Parvulins on HBV Replication

### 2.1. Pin1

Pin1, the protein product of the *PIN1* gene, is a conserved enzyme that functions as a key regulator of cell cycle progression, cell proliferation, transcription, and tumorigenesis by binding to phosphorylated Ser/Thr-Pro (S/TP) motifs in numerous substrates [12,29,30,31]. Homologs of Pin1 are conserved from *E. coli* to many unicellular eukaryotes and all multicellular organisms [16,17,18,19]. Pin1 consists of 163 amino acids and contains WW binding, flexible linker, and isomerase domains (Figure 1) [32,33]. The N-terminal WW binding domain binds to specific substrates that are phosphorylated at S/TP motifs, and the C-terminal isomerase domain catalyzes *cis/trans* isomerization of the bound peptide bond, inducing a conformational change [12,29,30,31]. These post-phosphorylation modifications enable Pin1 to regulate the stability, dephosphorylation status, transactivation, and subcellular localization of proteins, as well as protein functions such as transcriptional or catalytic activity [12,29,30,31,34,35,36]. The expression level of Pin1 is low in most normal tissues [37] but is enhanced in many human cancers [38]. Pin1 interacts with phosphorylated SP motifs in HBx to promote transactivation of HBx and hepatocarcinogenesis [24]. In addition, Pin1 binds to HBc via specific phosphorylated ^160^TP^161^ and ^162^SP^163^ motifs and stabilizes the protein in a phosphorylation-dependent manner to promote efficient HBV propagation [25]. In addition to HBx and HBc, Pin1 can bind to several other viral proteins to enhance viral replication and/or oncogenesis [36,39].

### 2.2. Par14 and Par17

*PIN4* is located on chromosome Xq13.1 and encodes two proteins, Par14 (13.8 kDa/131 amino acids) [20,21] and Par17 (16.6 kDa/156 amino acids), via alternative transcription initiation [22] (Figure 1). The *PAR17* mRNA accounts for only approximately 1% of the total parvulin mRNAs in various human tissues [22]. Correspondingly, the Par17 protein level is very low in human HCC cell lines compared to Par14 protein [27]. Unlike Par14, which is found in all multicellular organisms and many unicellular eukaryotes [16], Par17 is expressed in humans and great ape species only (Hominidae) [22,40]. Par14 is thought to function as a Pin1 complementing enzyme in cell cycle regulation, chromatin remodeling, metabolism, and signal transduction [41,42,43,44]. The overlapping cellular functions of Par14 and Par17 include protein folding, DNA binding, chromatin remodeling, cell cycle progression, ribosomal RNA processing, and tubulin polymerization [17,45,46,47]. According to the Human Protein Atlas, *PIN4* mRNA expression is highest in the liver [48], and Par14/Par17 protein levels are higher in several human HCC and HBV-replicating cell lines than in THLE-2 immortalized human liver epithelial cells [27]. Indeed, the involvement of Par14/Par17 in HBV replication has been demonstrated [27,28].

Whereas Pin1 has an N-terminal WW binding domain, Par14 and Par17 contain unstructured, highly post-translationally modified N-terminal basic residues [41,49] (Figure 1). This shared N-terminal basic domain shows 45% homology to the region surrounding the chromatin unfolding domain of HMGN proteins, which binds to nucleosomal DNA [49,50]. The additional 25 amino acids in the N-terminal extension of Par17, which are not present in Par14, form an N-terminal amphipathic α-helix [22,46] that functions as a mitochondrial targeting sequence [40]. Par14 and Par17 have basically identical functions [27,28]; however, one study demonstrated that the additional 25 N-terminal amino acids of Par17 are required for its Ca^2+^-dependent interaction with calmodulin and subsequent suppression of tubulin polymerization [51].

The PPIase domains of human Par14/Par17 and Pin1 share 39% sequence identity and greater than 50% homology [43,49]. The secondary structures of the PPIase domains exhibit a typical alpha helix and beta pleated sheet secondary element order (β_1_α_1_α_2_hβ_2_α_3_β_3_β_4_, with h representing a small helix or helical turn) [17,43,49]. Nuclear magnetic resonance spectroscopy revealed that the three-dimensional structure of human Par14 shows a high degree of conservation with the PPIase domain of human Pin1, which belongs to the FKBP superfold family [49]. The structure of the Par17 PPIase domain is expected to be identical to that of Par14. The surface topologies of Pin1 and Par14 differ [49], as reflected by the different substrate preferences of these proteins.

The different subcellular localizations of Par14 and Par17 proteins may be an important feature of their specific roles. Several studies have reported that Par14 is localized in the cytoplasm, nucleus, nucleolus, and spindle apparatus [41,47,50], whereas Par17 is localized in the cytoplasm [46,51] but is also targeted to the mitochondrial matrix by its mitochondrial targeting sequence in a membrane-potential and time-dependent manner [40]. Saeed et al. [27] reported that both Par14 and Par17 are localized in the nucleus, cytoplasm, and mitochondria, although this study only analyzed proteins at the mitochondrial surface and did not extend to the mitochondrial matrix. The same group reported that Par14 and Par17 can be associated with nuclear double-stranded DNA [27,28].

The double-stranded DNA-binding ability and subcellular localization of Par14 are regulated by post-translational modification of its N-terminal domain [41,50]. Ser19 to Ala (S19A) dephosphorylation mimetic mutation of Par14 alters the intracellular localization of the protein from predominantly nuclear to cytoplasmic [41]. In addition, a phosphorylation mimetic Ser19 to Glu (S19E) mutant of Par14 displays reduced DNA-binding affinity and is localized around the nuclear envelope but does not penetrate into the nucleoplasm [41]. Although Par17 is able to bind to double-stranded DNA under physiological salt conditions [40], the DNA-binding abilities of the corresponding dephosphorylation mimetic S44A mutant and phosphorylation mimetic S44E mutant have not been examined. However, Saeed et al. [27] found that the HBV transcriptional activities of the dephosphorylation mimetic S19A Par14 mutant and S44A Par17 mutant were higher than those of the phosphorylation mimetic S19E Par14 mutant and S44E Par17 mutant, suggesting that decreased DNA binding caused by S19/44 phosphorylation might affect the transcription promoting activity of Par14/Par17 [27]. This finding also indicates that the S44 residue of Par17 has the same role as the S19 residue of Par14 [27,28] and can be post-translationally modified via phosphorylation/dephosphorylation.

Mutation studies revealed that the negatively charged Glu46 (E46) and Asp74 (D74) residues of Par14 and E71 and D99 residues of Par17 contribute to substrate-binding specificity for positively charged amino acids preceding Pro [27,28]. Correspondingly, D99A mutation reduces the PPIase activity of Par17 [46]. Consistent with its preference for substrates containing Arg-Pro (RP) motifs [21], highest catalytic activity of Par14 was found toward substrates in which Pro was preceded by positively charged Lys (K) or Arg [52]. Recent studies have demonstrated that Par17 and Par14 share substrate-binding preferences [27,28]. One of these studies found that the RP motif of HBc is important for the interaction of Par14/Par17 with HBc and/or HBV core particles, and is preferable to the KP motif [28]. In other studies, no catalytic activity of Par14 was observed when Pro was preceded by the positively charged amino acid His [21,52]. Consistent with this finding, an R133H HBc mutant showed no Par14/Par17 binding and significantly reduced HBV replication [28].

## 3. Interaction of Par14/Par17 with HBV Proteins

### 3.1. Interaction of Par14 and Par17 with HBx

As mentioned above, Par14 and Par17 have substrate-binding specificity for positively charged amino acids preceding Pro [21,52], and HBx has two RP motifs (^19^RP^20^ and ^28^RP^29^) in its N-terminal regulatory domain [27]. This domain is thought to be involved in the interaction of HBx with multiple cellular proteins and signal transduction pathways [5]. The HBx ^19^RP^20^ motif was conserved completely, and the ^28^RP^29^ motif was highly conserved among isolates from 10 HBV genotypes (29 isolates from A to J genotypes) through amino acid sequence alignment of HBx [27]. In addition, co-immunoprecipitation analyses demonstrated that both RP motifs serve as Par14/Par17-binding sites [27], suggesting that the interaction with Par14/Par17 might facilitate isomerization of HBx and the induction of a structural rearrangement that stimulates multiple functions. Consistent with reports suggesting that negatively charged E46 and D74 of Par14 and D99 of Par17 contribute to substrate-binding specificity for positively charged amino acids preceding Pro [46,49], co-immunoprecipitation experiments using wild-type (WT) Par14 or Par17 and mutant variants in which E46/71 and/or D74/99 were mutated to Ala demonstrated that these residues are indeed crucial for the interaction of Par14 and Par17 with HBx [27].

A recent study found that knockdown (KD) of *PIN4* in HepG2 cells expressing HBx reduced the half-life of HBx significantly from 39 to 18 min [27]. By contrast, in cells overexpressing WT Par14 or WT Par17, the half-life of HBx was increased to 50–55 min, demonstrating that HBx is stabilized by Par14/Par17. HBx interaction-deficient Par14 and Par17 mutants (Par14: E46A/D74A; Par17: E71A/D99A) were unable to stabilize HBx in *PIN4*-KD cells. Similarly, even in the presence of exogenous WT Par14 or WT Par17, the stability of an interaction-deficient HBx mutant (HBx-AAAA) was similar to that of WT HBx in *PIN4*-KD cells. These findings suggest that interaction with Par14/Par17 enhances the stability of HBx, enabling it to perform multiple functions at various cellular sites [27].

Par14, Par17, and HBx are localized in the nucleus, cytoplasm, and mitochondria [20,27,40,53]. Co-immunoprecipitation of subcellular fractions and subsequent confocal microscopy analyses revealed that HBx–Par14 and HBx–Par17 interactions take place in all three of these cellular compartments, highlighting the multiple roles of HBx in the HBV life cycle [27]. Unlike WT HBx [27,53], the interaction-deficient HBx-AAAA mutant is localized mainly in the cytoplasm, even in the presence of Par14 or Par17, demonstrating that interaction of the RP motifs of HBx with Par14/Par17 facilitates efficient translocation to the nucleus and mitochondria [27].

Par14 and Par17 are non-histone DNA-binding proteins with chromatin remodeling properties [40,45,50]. A recent study demonstrated that endogenous Par14 and Par17 are recruited onto HBV cccDNA, and overexpression of these proteins upregulates HBV transcription [27]. The phosphorylation mimetic S19E Par14 and S44E Par17 mutants have lower cccDNA-binding affinities than the WT proteins [27], demonstrating that dephosphorylation of these residues of Par14 and Par17 is critical for their association with HBV cccDNA (Figure 2). Although HBx does not bind directly to DNA, it binds to various transcription factors and co-factors, and promotes the transcriptionally active state of HBV cccDNA in the presence of DNA-binding proteins, activating histone modifiers, and chromatin remodelers [4,5,10,54]. The negatively charged E46 and D74 residues of Par14 and E71 and D99 residues of Par17 bind to the HBx RP motifs, strengthening their binding affinity for cccDNA and inducing a transcriptionally active open chromatin state (Figure 2) [27]. Consequently, binding of Par14 and Par17 to cccDNA and transcriptionally active chromatin is diminished in the absence of HBx [27]. Par14 and Par17 may act as a bridge between HBx and cccDNA to promote the transcription of HBV RNA, resulting in upregulation of HBV DNA synthesis and ultimately increasing the level of cccDNA (Figure 2) [27]. In the absence of HBx, Par14 and Par17 may bind weakly to the transcriptionally inactive (closed) chromatin state of cccDNA [27]. Overall, in addition to HBx–Par14/Par17 interactions in the cytoplasm and mitochondria, HBx–Par14/Par17–cccDNA interactions in the nucleus (Figure 2) stimulate multiple stages of HBV replication [27].

### 3.2. Interaction of Par14 and Par17 with HBc and Core Particles

Like the conserved ^19^RP^20^ and ^28^RP^29^ motifs of HBx [27], HBc has a single ^133^RP^134^ motif that is conserved completely among ten human HBV genotypes [28]. The mammalian and avian HBc RP motif is conserved completely, and avian HBc has an additional conserved KP motif [28]. The HBc RP motif is located in the irregular Pro rich loop 6 (^128^TPPAY***RP***PN^136^) that follows helix α5 (amino acids 112–127) [55]. Loop 6 is highly conserved among ten genotypes of human and mammalian hepadnaviruses [28]. In this loop, Tyr132 (Y132) mediates the HBc dimer–dimer interaction to facilitate core particle assembly [55,56].

Par14 and Par17 co-immunoprecipitated with an HBV core particle assembly-defective dimer-positive HBc-Y132A mutant [56], and native agarose gel electrophoresis and immunoblotting revealed a physical interaction of HBV core particles with Par14/Par17 [28]. These findings suggest that Par14/Par17 can bind to both HBc and HBV core particles. Heat treatment at 65 °C weakened the interaction of core particles with Par14/Par17, indicating a dynamic on–off interaction [28]. Analyses of heat-treated and opened-up core particles [57,58,59] demonstrated that Par14 and Par17 can bind both outside and inside of the particles [28]. A cryo-electron microscopy analysis indicated that the CTD of HBc shuttles between the interior and exterior of the core particle [60], due in part to differences in its charge balance [61]. It is possible that the CTD of HBc that shuttles between the interior and exterior of the core particle [60,61,62] starts from loop 6 which includes the surface-exposed late domain such as ^129^PPAY^132^ motif [63] and ^133^RP^134^ motif [28] and extends to the structurally disordered HBc CTD. Native agarose gel electrophoresis and co-immunoprecipitation analyses revealed that, like the RP motifs of HBx [27], the RP motif of HBc and/or core particles are bound by the negatively charged substrate-binding E46 and D74 residues of Par14 and E71 and D99 residues of Par17 [28].

Analyses of several HBc RP motif mutants revealed that a positively charged Arg or Lys residue at position 133 (R/K133) is critical for core particle assembly, but P134 and P135 are not [28]. When R133 was mutated to Asp (D) or Glu (E), HBc showed an assembly-defective phenotype similar to that of HBc-Y132A [56], suggesting that a negatively charged residue at position 133 interferes with core particle assembly [28]. Due to a deficiency of interdimeric interactions, the dimer-positive HBc-Y132A mutant [56] does not form a high molecular weight HBc complex in non-reducing conditions [28]. Under the same conditions, the HBc-R133D and -R133E mutants do not form high molecular weight complexes, suggesting that dimeric HBc-R133D and -R133E are defective in core particle assembly due to a deficiency of interdimeric interactions [28]. The R/K133 residue of HBc is also critical for Par14/Par17 binding, and interaction of HBc with Par14/Par17 may stabilize core particles or improve the efficiency of their assembly [28]. Core particles are found in the cytoplasm, and core particle-Par14/Par17 interactions likely occur in this region (Figure 2) [28]. Notably, in a recent study, core particles assembled by HBc-R133A, -R133L (Leu), -R133H, -AAP, and -AAA migrated rapidly in an agarose gel [28], suggesting the formation of unstable particles [64].

As seen for HBx [27], the half-lives of HBc and core particles were decreased from >24 h to 20 h and from >24 h to 21 h, respectively, in *PIN4*-KD cells, and exogenous expression of Par14 or Par17 in these cells increased the stabilities of both substrates [28]. Upon mutation of the RP motif (HBc-AAP), the half-lives of HBc and the core particle were decreased further from 20 h to 12 h and from 21 h to 13 h, respectively [28], demonstrating the importance of the RP motif for the stability of HBc and core particles. Unlike WT HBc and core particles, HBc-AAP and core particles formed in this mutant were not stabilized by the addition of exogenous Par14 or Par17 due to the lack of interaction with these proteins [28]. Notably, Par14/Par17 increased the stability of HBc-Y132A but not that of HBc-R133D [28]. Overall, these findings indicate that HBc and the core particle are stabilized through specific interactions with Par14/Par17, mediated via the RP motif of HBc and the substrate-binding E46/71 and D74/99 residues of Par14/Par17 [28]. This stabilization effect likely strengthens the structural and regulatory roles of HBc and the core particle in HBV replication.

Par14 and Par17 also strengthen the regulatory role of HBc by enhancing its recruitment onto cccDNA [28]. Like the HBx–Par14/Par17–cccDNA interaction [27], interactions between HBc, Par14/Par17, and HBV cccDNA occur in the nucleus (Figure 2) [28]. In recent studies, overexpression of Par14/Par17 enhanced the recruitment of HBc, RNA polymerase II, and acetylated H3 onto HBV cccDNA [27,28], inducing transcriptional activation. In addition, KD of *PIN4* reduced the recruitment of these factors onto cccDNA [28]. The substrate-binding E46 and D74 residues of Par14, E71 and D99 residues of Par17, DNA-binding S19 residue of Par14, and S44 residue of Par17 are important for HBc recruitment onto cccDNA [28]. Recruitment of HBc-AAP onto cccDNA was lower than that of WT HBc, indicating that the ^133^RP^134^ motif of HBc is important for its interaction with Par14/Par17 and cccDNA [28].

The HBc CTD contains highly basic residues (Arg-rich, protamine-like) that resemble histone tails and are critical for non-specific nucleic acid binding [8,59,64,65,66,67,68,69]. Therefore, the HBc CTD binds to cccDNA, and the HBc RP motif interacts with Par14/Par17 [28]. CTD-deficient HBc lacks a nuclear localization signal and therefore cannot localize in the nucleus. In a recent study, when both CTD-deficient HBc [64] and WT HBc proteins were present [70], the recruitment of WT HBc onto cccDNA was reduced in the presence or absence of Par14/Par17, indicating that CTD-deficient HBc might interfere with the nuclear localization of WT HBc via an unknown mechanism [28]. In the same study, overexpression of Par14/Par17 upregulated the recruitment of WT HBc onto cccDNA, suggesting that both the CTD and RP motifs of HBc are critical for its recruitment onto cccDNA [28].

Overall, the findings described above indicate that the E46/D74 and E71/D99 residues of Par14/Par17 bind to the ^133^RP^134^ motif of HBc, the S19/44 residues of Par14/Par17 bind to cccDNA [27,28], and the CTD of HBc binds to cccDNA [71] (Figure 2). Therefore, HBc–Par14/Par17–cccDNA, HBc–cccDNA–Par14/Par17, and Par14/Par17–HBc–cccDNA interactions may occur in the nucleus (Figure 2) [28]. In addition to HBx–Par14/Par17–cccDNA interactions in the nucleus and HBx–Par14/Par17 interactions in the cytoplasm and mitochondria [27], triple interactions of HBc, Par14/Par17, and cccDNA in the nucleus and core particle–Par14/Par17 interactions in the cytoplasm can enhance HBV replication through increased transcriptional activity, increased core particle assembly and/or stability, and increased HBV DNA synthesis (Figure 2) [28].

### 3.3. Interaction of Par14 and Par17 with HBs and HBV Polymerase

An analysis of ten HBV genotypes revealed that HBs contain a highly conserved KP motif (LHBs: ^304^KP^305^; MHBs: ^196^KP^197^; SHBs: ^141^KP^142^; numbering according to the ayw HBV subtype), and HBV polymerase has five highly or completely conserved R(K)P motifs in its reverse transcriptase and RNase H domains (unpublished data). Consistent with their ability to bind to conserved RP motifs, Par14 and Par17 interact with HBV virions and HBs subviral particles, possibly via their conserved KP and R(K)P motifs, and overexpression of Par14/Par17 increases the secretion of virions and HBs subviral particles [27]. Since HBV polymerase has conserved R(K)Ps motifs, the possibility that it interacts with Par14/Par17 to affect HBV DNA synthesis should be investigated in future studies.

## 4. Summary of the Role of Par14 and Par17 in the HBV Life Cycle

As mentioned above, overexpression of Par14/Par17 upregulates HBV replication at multiple stages, from HBV cccDNA formation to the synthesis of RNA, DNA, and virions [27,28]. These effects were only seen in the presence of HBx [27,28]. The multiple subcellular localizations of HBx and Par14/Par17 in the nucleus, cytoplasm, and mitochondria [20,27,40,53] may be an important feature of their multifunctional roles in the HBV life cycle [5,27,28]. Luciferase, chromatin immunoprecipitation, and northern blotting assays revealed that overexpression of Par14/Par17 in HepG2 cells induced transcriptional activation and upregulated the levels of several HBV RNAs, including pgRNA and the subgenomic S and X mRNAs encoding HBs (LHBs, MHBs, and SHBs) and HBx, respectively [27,28]. These changes resulted in upregulation of the HBc protein level, core particle assembly, and HBV DNA synthesis [27,28].

As mentioned above, Par14/Par17 bind both outside and inside of core particles, and pgRNA encapsidation in core particles is enhanced via the interaction of Par14/Par17 in the particles with the RP motif of HBc [28]. The elevation of HBV replication caused by overexpression of Par14/Par17 may promote the recycling of relaxed-circular DNA, leading to an increase in the level of cccDNA and vice versa [27]. Overexpression of Par14 or Par17 in HepAD38 cells increased the extracellular (secreted) levels of HBs, HBc, HBs subviral particles, naked core particles, and virions [27]. The levels of Par14 and Par17 themselves were also increased in the cell supernatants, indicating that they are incorporated into virions and/or core particles [27,28]. Native agarose gel immunoblotting confirmed that Par14 and Par17 bound virions, HBs subviral particles, and HBV naked core particles [27]. In addition, in situ nucleic acid blotting revealed significantly higher levels of HBV DNA in virions and naked core particles following overexpression of Par14/Par17 [27].

As mentioned above, each residue in the ^19^RP^20^ and ^28^RP^29^ motifs of HBx and the ^133^RP^134^ motif of HBc is important for Par14/Par17-mediated upregulation of HBV replication [27,28]. The S19, E46, and D74 residues of Par14, and the S44, E71, and D99 residues of Par17, are important for Par14/Par17-mediated upregulation of HBV replication [27,28].

KD of *PIN4* in cultured liver cell lines reduced the levels of cccDNA, pgRNA, sgRNAs, and HBc, as well as core particle assembly and HBV DNA synthesis [27,28]. Likewise, treatment of the cell lines with parvulin inhibitors such as juglone (5-hydroxyl-1,4-naphthoquinone), a competitive irreversible inhibitor [72,73], and PiB (1,3,6,8-tetrahydro-1,3,6,8-tetraoxo-benzo[lmn] [3,8] phenanthroline-2,7-diacetic acid, 2,7-diethyl ester), a competitive reversible inhibitor [42], reduced HBV replication significantly [27]. These findings strengthen the evidence that Par14/Par17 is important for HBV replication [27,28].

As mentioned above, overexpression of Par14/Par17 enhanced the recruitment of HBx, HBc, RNA polymerase II, and acetylated H3 onto cccDNA [27,28], indicating transcriptional activation (Figure 2). These changes might trigger the unwinding of cccDNA from the closed to open conformation, possibly via the chromatin remodeling ability of Par14 and Par17, thereby promoting HBV transcription, HBV DNA replication, and virion secretion.

## 5. Conclusions and Perspective

Northern blot analyses have demonstrated high expression levels of Par14/Par17 in many tissues, including the heart, liver, skeletal muscle, placenta, kidney, and pancreas [21]. Similarly, according to the comprehensive Human Protein Atlas [48], Par14/Par17 is expressed at high levels in the liver, gastrointestinal tract, urinary tract, kidney, gallbladder, and endocrinological and male reproductive systems. Consequently, Par14/Par17 and the *PIN4* gene likely have essential biological roles. However, with the exception of HBV proteins, there is currently no evidence that other viral proteins interact with Par14/Par17 to affect viral replication and pathogenesis. Parvulin inhibitors such as juglone [72,73] and PiB [42] reduce HBc protein expression, core particle formation, and HBV DNA synthesis in HBV-transfected, HBV-replicating stable, and HBV-infected cells [27]. In addition, these inhibitors weaken the HBV core particle–Par14/Par17 interaction, indicating a dynamic on–off interaction [28]. Several parvulin inhibitors can inhibit HBV replication at various stages from HBV RNA to DNA synthesis (unpublished data). However, those inhibitors are not Par14/Par17-specific, such as juglone and PiB (unpublished data). The identification and characterization of specific parvulin inhibitors and/or Par14/Par17 inhibitors may lead to the development of new treatment options for CHB. Alternatively, KD of *PIN4* to inhibit HBV replication [27,28] could also be explored as a therapeutic approach. There are no therapeutic strategies utilizing Par14 and Par17 as targets. If possible, finding the Par14/Par17-specific small molecules which can be used for animal and human research would be of great benefit for CHB research to find a cure. Unfortunately, there is no research regarding the influence of the current treatment on the regulation of Par14 and Par17. Still, many things need to be done to understand the effects of *PIN4* (Par14 and Par17) on HBV replication.

## Figures and Tables

**Figure 1 viruses-15-00457-f001:**
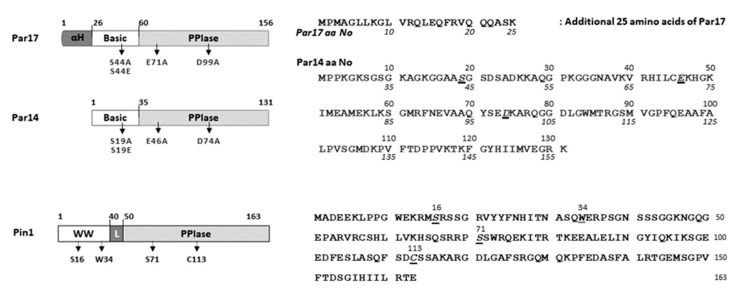
Schematic overview and amino acid sequences of the human PPIases Par17, Par14, and Pin1.

**Figure 2 viruses-15-00457-f002:**
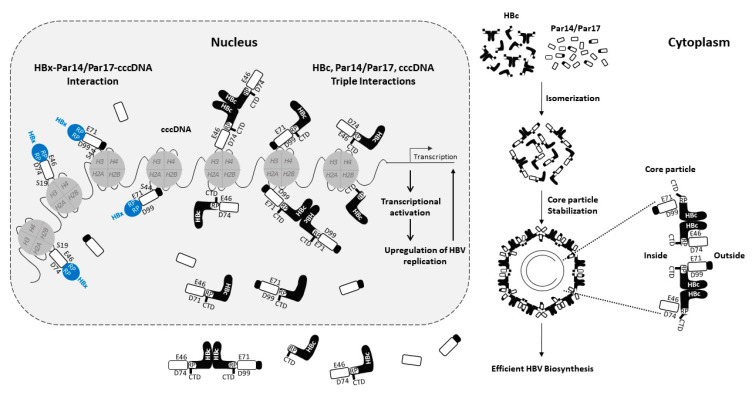
Par14 and Par17 upregulate HBV RNA transcription and DNA synthesis, thereby increasing the HBV cccDNA level. The HBV cccDNA minichromosome is associated with histone and non-histone cellular and viral proteins, including HBc and HBx. The HBx–Par14/Par17–cccDNA complex in the nucleus upregulates HBV RNA transcription, eventually stimulating multiple steps of HBV replication. HBc binds to cccDNA and Par14/Par17 through its C-terminal domain (CTD) and RP motif, respectively. Triple interactions between Par14/Par17, HBc, and cccDNA in the nucleus, and the interaction of Par14/Par17 with core particles in the cytoplasm, upregulate HBV replication. Par14 and Par17 bind directly to HBV cccDNA via their S19 and S44 residues, respectively. The substrate-binding E46 and D74 residues of Par14, and E71 and D99 residues of Par17, are critical for the interactions of these proteins with the RP motifs of HBx, the RP motif of HBc, and core particles. These residues also promote recruitment of HBc and HBx onto cccDNA (modified from [27,28]).

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
