# Peer review of "PPIases Par14/Par17 Affect HBV Replication in Multiple Ways"

_viruses, 2023, doi:10.3390/v15020457_

Round 1

Reviewer 1 Report

In this review, Dr. Kim has investigated the interaction of prolyl isomerases, Par14 and Par17, with HBV core (HBc) and X (HBx) proteins. The emphasis of this review is two papers, references 27 and 28, by Dr. Kim.  These papers each show evidence of interaction of the two HBV proteins with PAR14 and PAR17.  The presence of the prolyl isomerases seems to be to the viruses advantage and they do bind to motifs in HBc and HBx. However, the papers do not show that these interactions are required and do not exclude the possibility that the effects of the prolyl isomerases are indirectly related to the virus.  In HBc, the proline motif is always trans and the in vitro studies show no requirement for PAR14 or 17. In ref 27 and the review, the authors point out that the proline of the RP motif is not required for function. The review does provide a detailed description of PAR14 and PAR17 features and shows that their distribution correlates with HBx and HBc.

While generally well written, there are some difficulties.

In the abstract, the terminology for residues (e.g. S19 and E46/D74) is unclear. This is explained later in the paper. Simply referring to residues S19, E46, and D74 without the unexplained / is sufficient.

Several papers, particularly biochemical, are misdescribed. For example, the electron microscopy in reference 57 was transmission EM (cryo-TEM), not scanning EM, and it did not show that the CTD could shuttle between the interior and exterior of the core particle

Author Response

Responses to the reviewer 1

Thank you for your thorough review.

  1. …the papers (1-1) do not show that these interactions are required and (1-2) do not exclude the possibility that the effects of the prolyl isomerases are indirectly related to the virus.

1-1.

Yes. I do not show and claim that these interactions are required.

Without their interaction, virus would be dead that is required, but through their interactions, virus proteins (HBx and HBc) and core particles are stabilized, and HBV replication was enhanced. And without their interactions, virus proteins and core particles are less stabilized, and HBV replication was decreased. These results demonstrate that the importance of their interaction.

So, I have shown that these interactions are important for HBV replication from abstract throughout the manuscript. Since this is a review paper, I did not show the specific results, but I described throughout the manuscript.

NAGE and immunoblotting with anti-PIN4 show the physical interaction with Par14/Par17 and core particle (Refs 27 and 28). Co-immunoprecipitation, SDS-PAGE, and immunoblotting with anti-PIN4 and anti-HBc show the physical interaction with Par14/Par17 and HBc (Ref 28). Co-immunoprecipitation, SDS-PAGE, and immunoblotting with anti-3×FLAG (Par14 and Par17) and anti-Myc (HBx) show the physical interaction with Par14/Par17 and HBx (Ref 27, Fig. 7 A, B, D). Colocalization of HBx WT and Par14 WT or HBx WT and Par17 WT by confocal microscopy results further support the result.

Since S19A, S19E, E46A, D74A mutants and E46A/D74A double mutants of Par14 and S44A, S44E, E71A, and D99A mutants, E71/D99 double mutants of Par17 failed to rescue HBV replication, for Par14/Par17-mediated upregulation of HBV replication (Refs 27 and 28).

With HBx RP motif mutants, HBV replication cannot be increased in the presence of Par14 or Par17 (Ref 27, Fig. 9).

1-2.

Yes, I agree.

Even though I demonstrated direct effects of Par14 and Par17 on HBV, I cannot exclude the possibility that the effects of the Par14 or Par17 are also indirectly related to the virus. This can be studied further. Par14 and Par17 are known to bind several host proteins (eg., tubulin) and this can also affect to the virus.

Before our publication we did not know that Par14 or Par17 have any effects at all to the virus. We are getting to know some relationship with Par14/Par17, HBV and several indirect factors, maybe.

  1. …. terminology for residues (e.g. S19 and E46/D74) is unclear.

Thank you for your comments.

As you recommended, I changed S19 to Ser19 (S19), E46/D74 to Glu46/Asp74 (E46/D74) in the abstract and in the manuscript when they appeared first.

Now it would be little bit easier to understand.

Thank you.

  1.  Several papers, particularly biochemical, are misdescribed. For example, the electron microscopy in reference 57 was transmission EM (cryo-TEM), not scanning EM,

Thank you for your correction. I really appreciate that.

Through a manuscript, I mistakenly put as cryoscanning EM. Refs 57 did not mention cryo-TEM through the manuscript (I think you are right since I am not an expert in structural biology).

However, Ref 62 mention that the Acknowledgement that ‘Microscopy data were collected at the IU Cryo-Transmission Electron Microscopy Facility…. ’. Mention that their cryo EM method was cryo-TEM. However, they also mention as cryo EM through the manuscript, so I changed as cryo EM , instread of cryo TEM.

  1. …it did not show that the CTD could shuttle between the interior and exterior of the core particle.

Yes. They did not use the word shuttle and exactly say that CTD could shuttle between the interior and exterior of the core particle.

However, Ref 57 (Yu et al., PLoS ONE 2013) they did say that “…. Weak densities emanate from the rims of positively charged channels through the icosahedral three-fold and local three-fold axes. We attribute these densities to the exposed portions of some ARDs, thus explaining ARD’s accessibility by proteases and antibodies. Our data supports a role of ARD in mediating communication between inside and outside of the core during HBV maturation and envelopment.”

Ref 58 (Selzer et al., JBC 2015) mentioned that “Adding negative charge to CTDs increases capsid stability and decreases CTD exposure.” “….quasi-equivalent CTDs exhibit different rates of exposure…..” “…. A structural role for CTD phosphorylation and indicate crosstalk between CTDs within a capsid particle.” “This also suggests a difference in CTD exposure between Cp183-EEE and Cp183-WT capsids. “There Are Two Populations of CTDs with Different Rates of Exposure…. For Cp183-WT capsids, 25% of CTDs had a slow half-life of 28.8 min, whereas 75% of CTDs were digested more quickly, having a half-life of 2.5 min….For Cp183-EEE capsids, 50% of CTDs were cut with a slow half-life of 89.6 min and 50% with a fast half-life of 1.5 min.” “Our results strongly suggest that HBV CTD phosphorylation can contribute to capsid stability and alter CTD accessibility.”

Also, Ref 62 (Wang et al., PLoS Pathogens 2012) mentioned that “We further provide direct evidence of partially exposed CTDs on the capsid exterior, suggesting how they may play a role in intracellular trafficking and secretion of HBV cores.”

From these notions, I described ‘CTD could shuttle between the interior and exterior of the core particle (Ref 28) citating Ref 57 and Ref 58’.

Reviewer 2 Report

In this review, combined with the published and unpublished data of the Professor Kyongmin Kim’s group, the author systematically summarizes the effects of host parvulins including Pin1, Par14 and Par17 on HBV replication, the interaction of Par14/Par17 with HBV proteins and the role of Par14/Par17 in HBV life cycle. From this review, we can find that Par14 and Par17 can regulate HBV replication from multiple levels and are the potential antiviral targets for HBV. Accordingly, the specific Par14/Par17 inhibitors may lead to the development of new treatment options for CHB. This review is well written and is of high quality. The references of this manuscript are very representative.

Author Response

Thank you for your generous review.

I really appreciate it.

Reviewer 3 Report

The present review summarized the mechanism underlying the effect of human parvulin 14 (Par14) and parvulin 17 (Par17) on the hepatitis B virus (HBV) replication. The review is overall comprehensive and well written. Some minor points are listed as below.

1. Are there therapeutic strategies utilizing Par14 and Par17 as targets?

2. What are the influence of the current treatment on the regulation of Par14 and Par17?

Author Response

Thank you for your kinf review.

  1. Are there therapeutic strategies utilizing Par14 and Par17 as targets?

Thank you for your interest.

Unfortunately, No. There are no therapeutic strategies utilizing Par14 and Par17 as targets. Our unpublished study reveals that several parvulin inhibitors can inhibit HBV replication in the transfected and infected cells, but those inhibitors are not Par14/Par17 specific, like Juglone and PiB. If possible, finding the Par14/Par17 specific small molecules which can be used for the animal and human research would be great benefits for CHB research to find a cure.

  1. What are the influence of the current treatment on the regulation of Par14 and Par17?

Unfortunately, there is no research regarding the influence of the current treatment on the regulation of Par14 and Par17. It would be interesting to check at least in the transfected and infected cells. Still many things need to be done to understand the effects of PIN4 (Par14 and Par17) on HBV replication.

I will incorporates above contents in the manuscript.